# Application of Internet of Things in a Kitchen Fire Prevention System

**Wei-Ling Hsu** [1], **Ji-Yun Jhuang** [2], **Chien-Shiun Huang** [3], **Chiu-Kuo Liang** [4] **and Yan-Chyuan Shiau** [5,*]

1   School of Urban and Environmental Science, Huaiyin Normal University, Huai'an 223300, China
2   Program of Technology Management, Chung Hua University, Hsinchu 30012, Taiwan
3   Department of Civil Engineering, Chung Hua University, Hsinchu 30012, Taiwan
4   Department of Computer Science and Information Engineering, Chung Hua University, Hsinchu 30012, Taiwan
5   Department of Landscape Architecture, Chung Hua University, Hsinchu 30012, Taiwan
*   Correspondence: ycshiau@ms22.hinet.net; Tel.: +886-916-047-376



**Featured Application: The results of this research can be used make an installation in the kitchen area of the home to improve the safety of cooking with gas.**

**Abstract:** When using a gas stove to cook, the food or oil in the pot or pan may catch fire because of an excessively high temperature. In addition, people may be uncertain as to whether they have turned off the gas stove before leaving home and therefore feel compelled to return home to check. To solve these problems, this study has developed a smart kitchen fire prevention system that possesses the following devices and functions. (1) Sensors are installed above the stove top. When they detect flames, high temperature, or a gas leak, they immediately activate the gas shutoff device to turn off the gas supply. (2) The alarm produces a loud sound and flashes to warn the residents. (3) The Line reporting system sends Line messages to notify the residents and the community management center, and the main entrance door is automatically unlocked to allow relevant personnel to enter the house to deal with the accident. (4) An Internet protocol camera is installed in the kitchen to enable the residents to monitor the gas stove on their mobile phones. If they find the gas stove is still turned on, they can activate the gas shutoff device to shut off the gas supply from their phones. The system developed in this study can effectively reduce the loss that results from a kitchen fire.

**Keywords:** flame detector; temperature sensor; gas sensor; gas shutoff device; Line reporting system

---

## 1. Introduction

### 1.1. Background

When a fire breaks out, it can often cause major loss of life and damage to property, due to the spread of the fire. Early detection of fire sources and effective countermeasures have become important issues in fire prevention and protection of life and property [1]. As building fires tend to increase the number of casualties and property damage, it is necessary to take appropriate action accordingly. There has been an increasing effort to develop such disaster management systems worldwide by applying information communication technology (ICT), and many studies have been conducted in practice. The core idea behind the smart city is to collaborate and integrate the information from urban areas and utilize ICT, which includes cloud computing, Internet of Things and so on. The main characteristics of smart city are scalability, interoperability, limited power consumption, fast deployment, multi-modal

access and robustness [2,3]. The need for a basic unified architecture for smart city based applications is not new, but a proper solution that can map real world smart city applications to the standard architecture is not currently available [4]. The smart building is also an important part of the smart city. The gas stove is a type of cooking device used in most households in Taiwan. Compared with an electric stove, it saves electricity and can cook food rapidly because it produces large flames. However, users or older adults may forget they are cooking or forget to turn off the stove because they are busy or have memory problems. Double-income couples may be unsure whether they have turned off the stove and therefore feel compelled to, during their commute to work, return home and check. These problems affect numerous households.

Business Insider, a British magazine, commissioned an online survey of 14,300 expats who represented 174 nationalities and lived in 191 countries to rate various aspects of life, including leisure options, personal happiness, travel, transport, health and well-being, and safety and security. The company published the ranking of top 21 countries with the best quality of life in January 30, 2017, and Taiwan was ranked first because of a comprehensive medical care system and affordable price level [5]. Among the aspects measured by the assessment, personal happiness, travel, and leisure options are related to food in Taiwan. Taiwanese people prefer to cook over a large flame and high heat; however, the oil in woks may catch fire because of the excessively high temperature that is produced, which could lead to kitchen fires. Therefore, devising a method to automatically shut off gas when a gas stove catches on fire is of critical importance to home safety.

### 1.2. Research Motivation and Objectives

In order to prevent stoves from catching fire because of a pot or pan boiled dry or heated over an extremely high temperature, this study has developed a smart kitchen fire prevention system that has the following main functions.

### 1.2.1. Kitchen Fire Prevention

(1)　A sensor is installed above the stove top. When it detects overly high flames or temperature or a gas leak, the automatic shutoff device will be activated to shut off gas supply to the stove.
(2)　A reporting system then reports the accident to the family members and the community management center via Line, a messenger app, and the main entrance door of the house in question is automatically unlocked to enable the emergency personnel to enter the house to extinguish the kitchen fire.

### 1.2.2. Remote Monitoring and Gas Shutoff

(1)　An Internet protocol (IP) camera is installed in the kitchen at a location and height that allows users to see the gas stove.
(2)　When users are unsure whether they have turned off the gas stove, they can check the live video of their cameras from their phones. When necessary, they can activate the gas shutoff device to shut off the gas supply by using voice control or clicking on an activation button on their phones, thereby ensuring home safety.

### 1.3. Problems to be Solved

To provide a complete kitchen accident prevention mechanism, problems that must be solved include the following:

(1)　Users may forget they have something cooking on the stove when they are busy. Therefore, when a pot or pan boils dry, a sensing element is required to detect the high temperature and issue light or sound alarms to warn residents and neighbors.

(2)   When a pot or pan catches fire because of an overly high temperature or a high alcohol content in food, the fire incident may become more serious if it is not handled appropriately. Therefore, an automatic gas shutoff device is necessary in such a situation.

(3)   When a household kitchen catches on fire, the owner must report the accident to the management center or emergency units immediately via a communication mechanism. However, if the main entrance door to the house is locked, the optimal time for rescue may be missed. Therefore, the kitchen fire prevention system automatically unlocks the door to enable others to enter the house for rescue operations.

(4)   When users are unsure whether they have turned off the stove, they can view the live video of the stove by using the remote monitoring system. If the stove is still turned on, a remote control system is required for the users to shut off the gas supply to the stove.

## 2. Literature Review

According to the Exploration of Causes of Housing Fires in Taiwan conducted by the Architecture and Building Research Institute, Ministry of the Interior [6], the total number of fire incidents in 1999 in Taiwan was 16,389, 36.1% of which were building fires, comprising 5,913 cases. In addition, among the building fire incidents, 61.3% of those were housing fire incidents, totaling 3,626 cases, indicating a high percentage of housing fire incidents among all fire incidents in Taiwan. The term "Smart Sensors", one of the most strategic devices of Micro System Technologies (MST), refers to sensors that contain both sensing and signal processing capabilities with objectives ranging from simple viewing to sophisticated remote sensing, surveillance, search/track, weapon guidance, robotics, perceptron and intelligence applications [7]. Many services and applications integrating these technologies into daily life have come to form an Internet of Things (IoT) [8].

### 2.1. The Development Trend of Smart Building

At present, most cities are faced with high-rise and dense buildings, so it is the most important task to prevent and solve any type of fire situation quickly and effectively, in order to most effectively reduce losses in terms of life and property [9].

The introduction of new technologies has provided us with many new options to achieve security in smart buildings and smart cities. The intelligent feature is that when such disasters occur, these buildings and cities should be able to implement emergency response measures quickly and holistically. Key technologies and factors that help prevent fires should also be noted. Some of the key technologies that can be used to achieve this ideal include:

(1)   Significant growth and easy access to Internet connectivity and bandwidth.
(2)   Ubiquity of Smart Phones and Tablets along with their inbuilt notification systems.
(3)   Advancement of wireless technologies, especially for IoT enabled sensors.
(4)   Economical access to Cloud-based Apps and data storage.
(5)   Adoption of Computer aided Facility Management (CAFM), Building Information Modeling (BIM) and virtual reality (VR) technologies for efficient operation and management of buildings.

### 2.2. Kitchen Fire

People suffering from a loss of autonomy caused by a cognitive deficit generally have to perform important daily tasks (such as cooking) using devices and appliances designed for healthy people, which do not take into consideration their cognitive impairment. Using these devices is risky and may lead to a fire tragedy [10]. Cognitively-impaired persons, such as elderly people with Alzheimer's disease or young people with brain injuries, suffer from a loss of autonomy. This deficit induced by their condition limits these individuals in performing their essential activities of daily living (ADL), such as bathing or cooking [11]. These people generally have to routinely perform tasks, in their home, using devices that do not necessarily take into consideration their cognitive deficit. These devices are

not adapted to their conditions, and they often come with unacceptable fire risks [12]. In order to successfully escape a fire and to douse the fire source, the fire must be detected at its initial stage. The installation of a fire alarm system is the most convenient way to detect a fire early and avoid losses. Fire alarms consist of different devices working together that have the ability to detect fire and alert people through visual and audio appliances. The detection devices (i.e., heat, smoke, and gas detectors) detect events and activate the alarm automatically, or sometimes the alarms are activated manually. The alarm may consist of bells, mountable sounders, or horns [1].

According to the fire statistics released by the National Fire Agency [13], cooking is the second leading cause of building fires (7,579 incidents), with only electrical appliances causing more fires from 2013 to 2018 (Table 1). From 2017 onward, the statistical data indicate a considerable change. This is because the National Fire Agency implemented a new fire categorization system in 2017 to clearly present the type of fires, which facilitates fire policy formulation in accordance with international standards. Specifically, fire incidents are categorized into A1, A2, and A3. A1 refers to fire incidents leading to deaths; A2 refers to fire incidents that lead to injuries, involve legal disputes, are categorized as arson, or have causes requiring further investigation; and A3 refers to fire incident cases that are closed after a "fire rescue attendance form" is completed. In view of the high percentage of fires caused by stove-cooking, determining the adequate measures to adopt to prevent kitchen fires caused during cooking is pivotal to ensuring home safety.

**Table 1.** Number of fire incidents and fire losses in Taiwan from 2013 to 2018.

| Item \ Time | 2013 | 2014 | 2015 | 2016 | 2017 | 2018 | Total |
|---|---|---|---|---|---|---|---|
| Total | 1451 | 1417 | 1704 | 1856 | 30,464 | 27,922 | 64,814 |
| Arson | 210 | 213 | 268 | 278 | 323 | 285 | 1577 |
| Suicide | 19 | 27 | 21 | 22 | 72 | 64 | 225 |
| Lights and candles | 11 | 6 | 18 | 16 | 67 | 51 | 169 |
| Cooking | 63 | 69 | 72 | 125 | 3659 | 3591 | 7579 |
| God and ancestor worship and tomb sweeping | 42 | 43 | 45 | 31 | 1936 | 1604 | 3701 |
| Cigarette | 135 | 146 | 147 | 169 | 1461 | 1530 | 3588 |
| Electricity | 508 | 451 | 582 | 608 | 3433 | 2972 | 8554 |
| Mechanical equipment | 41 | 30 | 29 | 40 | 469 | 365 | 974 |
| Playing with fire | 12 | 12 | 13 | 16 | 57 | 39 | 149 |
| Fires started for warmth | 5 | 3 | 4 | 4 | 28 | 24 | 68 |
| Construction accidents | 35 | 42 | 38 | 51 | 272 | 283 | 721 |
| Spontaneous combustion of flammable products | 7 | 9 | 8 | 10 | 32 | 26 | 92 |
| Gas leak or explosion | 26 | 16 | 28 | 25 | 98 | 88 | 281 |
| Chemicals | 11 | 8 | 5 | 10 | 24 | 19 | 77 |
| Burning of firecrackers | 15 | 19 | 27 | 22 | 181 | 229 | 493 |
| Traffic accidents | 15 | 6 | 18 | 23 | 121 | 91 | 274 |
| Natural disasters | 1 | 3 | 2 | 2 | 19 | 7 | 34 |
| Unattended small sources of fire | - | - | - | 142 | 5810 | 6353 | 12,305 |
| Unknown causes | 17 | 25 | 17 | 23 | 48 | 32 | 162 |
| Other | 278 | 289 | 362 | 239 | 12,354 | 10,269 | 23,791 |

2.2.1. Three Steps for Extinguishing a Grease Fire

Many fire accidents start from the kitchen, and if such fires are not extinguished immediately, they result in irreparable harm, perhaps even tragedy. A common cause for kitchen fires is excessively hot oil catching on fire, which could be caused by overheating cooking oil or stew. Many people intuitively pour water onto the oil to put out the fire, but doing so causes the oil to splatter, making the situation more dangerous.

The boiling point of oil is generally higher than 100 °C, at which point water vaporizes. When water enters the oil that has caught on fire and vaporizes, the water vapor carries burning oil droplets into the air and spreads the fire. The correct three steps for extinguishing a grease fire are as follows [14]:

(1)    When the fire is only within the pot or pan, turn off the gas stove.
(2)    Cover the pot that is on fire with a lid because this puts out the fire by blocking its access to air.
(3)    If the fire has been put out but the temperature in the pot remains high, directly opening the lid may cause the fire to reignite. Therefore, leave the pot alone and let it cool down naturally.

Every kitchen should be equipped with a dry-powder fire extinguisher for emergency use. Installing home smoke alarms can enable residents to become aware of the fire early and respond to it in a timely manner.

2.2.2. Effective Strategies for Kitchen Fire Prevention

Fire is a fundamental element for cooking; with fire, people create their food culture. However, fire can lead to disaster and cause irreparable harm. Fire prevention is thus crucial, particularly in the kitchen where fire is frequently used. Various food businesses operate on the basis of different standards use different types of fuels in the kitchen, thereby having different kitchen facilities and cooking environments. With the continuous advancement of kitchen facilities and changes in the methods of using fire, numerous factors are present that may cause a fire in addition to inherently flammable nature of stoves, coal gas, diesel fuel, and liquefied petroleum gas. The following characteristics of a kitchen make it a potential place for fire accidents [15]:

(1)    Abundance of fuel

The kitchen is a place where fire is used for cooking. Commonly used fuels include liquefied petroleum gas, coal gas, natural gas, carbon, and alcohol, all of which are likely to leak, combust, and even explode.

(2)    Large amount of cooking oil fumes

A kitchen environment is typically humid, and products of incomplete combustion and cooking oil fumes produced from oil evaporation accumulate and form a thick layer of a combustible oily and powdery substance that attaches to walls and collects in air ducts. If the cooking oil fumes are not cleaned off immediately, they can cause fire easily.

(3)    Complex power lines for electrical appliances

A kitchen is usually a small space, but kitchens tend to be occupied by numerous large kitchen appliances, most of which are powered by electricity or involve the use of fire. Inappropriate use of these appliances in the usually humid kitchen can easily lead to a short circuit and further result in electric shocks or fire disasters.

2.2.3. Causes of Kitchen Fires

The causes of kitchen fires include the following:

(1)    Users putting too much oil in a pot when deep frying food; the excessive amount of oil may splash out of the pot when boiling and make contact with the cooking fire.

(2)　Heat cooking oil for an excessively long time when deep frying; an oil temperature exceeding 240 °C leads to spontaneous combustion of the oil.

(3)　Ignition of a gas stove with a hot pot placed improperly on it causes the combustible substances to catch fire.

(4)　Leaving stew unattended on a gas stove while cooking; oil that floats on the soup spills out of the pot and make contact with the cooking fire.

(5)　Because of the high humidity and temperature in the kitchen, a large amount of oil residue is attached to the kitchen surface and the plastic wire insulation layers are oxidized. In addition, the constant presence of smoke, dust, and oil residue in a kitchen may easily lead to a short circuit among electrical appliances, electrical kitchen facilities, lighting devices, and switches in the kitchen, which then causes a fire accident.

(6)　The excessive accumulation of oil and grease on the surface of the exhaust hood can draw flame up into the ventilation ducts when pan-frying, which usually creates a high flame, and result in fire.

### *2.3. Internet of Things*

Based on people's demand for energy saving in their homes, the development of smart house technology is an inevitable result [16–20]. A smart house refers to the integration of all automation equipment through the network system to provide a service that is highly efficient overall, thereby ensuring that the public's needs are met in terms of home safety, a healthy and humane living environment, convenience, and quality of life [21]. The modern smart home system includes the following subsystems: lighting, climate control, security, technical incidents prevention subsystem and others [22–24].

There are four common methods of IoT connection control: Internet Control (W-Fi), Area Network Control (WebSocket), Bluetooth and Serial Port. Each of these four methods has its own advantages. For example, to control US devices in Taiwan, you must use Wi-Fi control. If you want to control multiple devices at the same time, you can consider WebSocket and Bluetooth connection. If you only want simple wiring control, you can use the Serial Port approach.

The wireless sensor network, being a dominant prerequisite in the modern pervasive environment, has nodes connected with multi-hop to transmit and reinforce continuous monitoring with real-time updates from the field environment [25]. In this context, IoT will also play an active role by connecting and enabling devices to the Internet [26]. Most of the fire alarm systems use the technology of a wireless sensor network (WSN). WSNs have gained popularity because they have a variety of uses in different applications, such as target tracking [27,28], localization [29], healthcare [30,31], Smart Transpiration [32], environmental monitoring, and industrial automation [33]. A WSN is also used to automatically or manually collect data and security monitoring [34,35].

Most of the smartphone-based applications either employ crowdsourcing or crowd sensing, utilizing smartphone sensors of citizens. However, a few can also be found that takes input from other sensor devices (such as wearable sensors) but collect data through the smartphone interface of citizens. Here, in this work, we have considered an opportunistic crowd sensing based application [36]. The level of such system's intellectualization increases with time. Therefore, the practice of neuro controllers for the smart home system implementation is an important and urgent task [37–40].

The IoT enables various devices and appliances that are attached with micro sensor chips, such as radio-frequency identification tags, sensors, and wireless communication chips, to communicate and interact with each other through wireless and wired networks (communication and interactions between objects, between people and objects, and between people). In recent years, sensors have been widely used for fire detection [41–46]. The IoT provides management and service functions that coordinate comprehensive types of objects, achieve high efficiency, energy saving, safety, and environmental protection, and offer integrated services of management, control, and operation.

The key characteristics of an IoT setup include the following: (1) instrumentation, (2) interconnectivity, (3) intelligence, and (4) facilitation of smart life and services through instrumentation and networking technologies.

Teslyuk et al. have developed the technical structure of the accident prevention subsystem for the smart home system. In the development process, the subsystem model is built on the Petri network and neural network, and the physical model is built using the Arduino microcontroller. The subsystem was developed using this technical model, and software and hardware tools were also presented [20].

*2.4. Current Research on Kitchen Fire Prevention*

The security concerns of elderly people have become a significant issue around the world. The ratio of the elderly population is getting higher and higher. By 2036, the proportion of people above 65 in Canada will account for 23% to 25% of the total population, and by 2061 will reach 24% to 28% [47]. Japan is the country with the highest proportion of the world's elderly population. In 2013, the 65-year-old population accounted for 25.1% of the total population, and by 2050 it will reach 40% [48]. Many studies have established preventive methods to improve safety in the kitchens of elderly people. Proactively responding to hazards in the kitchen can prevent cooking-related hazards. Researchers have managed to use a culinary safety system consisting of perceptron nodes to process sensory data based on risk prevention algorithms to monitor and prevent dangerous events that may occur in the kitchen environment [49].

In this section, we mainly discuss and summarize the progress of IoT technology and the latest smart disaster prevention system in recent years. Due to advances in sensors and microelectronics, fire detection technology has made great progress in the past decade [50,51]. Advances in various emerging sensor technologies for fire detection and monitoring are explained by Liu and Kim [52]. Most fire detection techniques fall into two categories, one that uses visual analysis video and image processing techniques to detect fires, and the other that uses sensors to perform fire detection [53,54]. After the sensor detects various types of kitchen disaster signals, the system performs intelligent alarm notification, remote information notification, and automatically shuts off the gas supply. Many studies on gas leak safety systems have been published [17,55,56]. These systems use gas sensors to detect gas leaks, initiate alarm procedures, send relevant information to households, and take precautions to avoid accidents [57,58]. The Global System for Mobile Communications (GSM) module is used, which alerts the user by sending an Short Message Service (SMS) [59]. Past research has focused on the GSM system, but with the advancement of the technology, various communication software (such as WhatsApp, Line, and WeChat) has gradually replaced traditional communication modes. This study uses the "Line" communication software to replace the traditional SMS, and uses the smart phone's App to perform remote control.

2.4.1. Video Frames and Process Images Technologies

Traditional single-point heat and smoke detectors are widely used, and they are usually only responsible for detecting limited areas in space. In large rooms and high-rise buildings, smoke particles and heat can take a long time to be detected. In the detection of fires, we prefer to use video-based fire detectors because it can effectively perform large-scale room and high-rise building fire monitoring, and even perform outdoor fire detection needs [60,61]. Currently most video fire detections use the visual characteristics of fire, including color, texture, geometry, flicker and motion. [55,62] The gas detector can detect the sign of the disaster even before the fire occurs [56]. Several smoke detection studies have been published using cameras to detect smoke. Since the smoke is grey and translucent, the edges of the high frequency image frame lose clarity and become an indicator of smoke. The distinction between different types of smoke is through the examination of changes in background tones, smoke pixels, blurred backgrounds, and illumination segmentation [51,63]. Due to the risks associated with the use of video frame detectors, this study used a combination of sensors to detect fires.

### 2.4.2. Sensor-Based Fire Detection Technologies

The actuator provides a wide range of possibilities for human–machine interaction, including appropriate interventions for each detected risk situation, and an appropriate response based on user needs. Establishing a sensor-based safe cooking environment requires in-depth risk analysis [49]. The disaster risk of the kitchen consists of three factors: fire, burn and intoxication. The following is a summary of these three factors.

The sensor-based fire detection technology is highly accurate, easy to install, inexpensive, and easy to deploy [51]. The vision-based visual fire detection algorithms are used for fire color modeling and motion detection, and the sensor network can be combined with this system. These network combinations seem attractive, but they increase system overhead and increase the complexity of system installation and deployment [64]. Compared with the above techniques and methods, safe from fire (SFF) has a simple strategy for detecting fires, and can inexpensively and efficiently deal with fraudulent fire scenarios. It is also effective for early detection of fires. In this study, Arduino sensing components were used as fire detectors for home kitchens. These detectors include a flame detector, a temperature detector, and a gas detector. Information about the program and control modules will be discussed in detail in Section 3.

### 2.4.3. Related Patents

This study searched patents related to kitchen fire prevention on the Global Patent Search System and found the following two patents:

(1)　TW M501536 (Fire detection and extinguishing equipment) [65]

Name: Fire detection and extinguishing equipment. This kitchen fire detection and extinguishing equipment consists of natural gas pipelines, a gas stove, an exhaust hood, a flame detector, and an extinguishing device. The extinguishing device consists of a gas cylinder connected to a branch pipe through a solenoid valve; the branch pipe and its nozzle are installed on the exhaust hood. When the flame detector detects flames, it outputs a signal to the driver through a signal transmission device. After receiving the signal, the driver turns off the gas supply and opens the solenoid valve to discharge the fire extinguishing gas from the cylinder to the solenoid valve, branch pipe, and then the nozzle to extinguish the fire on the gas stove. The drive motor of the exhaust hood is automatically turned on, which activates the exhaust fan to clear the air. Thus, the equipment effectively suppresses fire accidents and improves safety in kitchens.

Comparison between patent TW M501536 and this study: The patented system and this study both use the same fire detection module and gas shutoff device. However, the patented system does not have an alarm device, an automatic door unlocking function, or a Line message reporting mechanism.

(2)　TW 201709157 (Kitchen fire prevention system and kitchen monitoring device) [66]

Name: Kitchen fire prevention system and kitchen monitoring device. The system can be installed on the gas stove and gas pipelines. When a fire accident occurs, it can send a message to a portable warning device to wirelessly notify users. The kitchen monitoring device starts to count time as it detects the user moving a certain distance away from the gas stove. The device sends warning messages to the portable warning device wirelessly at certain time intervals. As the device detects that the user has left the gas stove for a certain time period, it controls the solenoid valve and shuts off the gas supply. The kitchen monitoring device and portable warning device can repeatedly remind the user to check the gas stove after leaving the kitchen. Once the user leaves the gas stove for a particularly long time or particularly long distance, the device automatically shuts off the gas supply, thereby effectively preventing fire accidents. Thus, this is a convenient and feasible fire prevention system.

Comparison between TW 201709157 and this study: The patented kitchen monitoring system encompasses an automatic shutoff device, a wireless reporting mechanism, and an alarm system, all of

which are also proposed in this study. However, the patented system does not have flame, temperature, and gas detection and automatic door unlocking functions.

## 3. Research Design

### 3.1. Software and Hardware Used in this Study

For any IoT-based application, a common factor is the presence of various sensors which are actually sensing the environment [3]. The sensing elements employed in this study are introduced as follows:

(1)     Arduino

Arduino [67] is a technology with open authorization applied in an interactive development environment and is comprised of hardware, software, and expansion components. The hardware is a prefabricated control board, and the software development environment is an open source code, which can be downloaded from the Arduino official website. With programming language syntax similar to C/C++, Arduino possesses a text editing interface, a toolbar, a graphical control interface, and an error editor. We uploaded programs to Arduino, which facilitated communication between various sensors. The console interface can record the complete execution messages and can be used to monitor the Arduino I/O values (Figure 1).

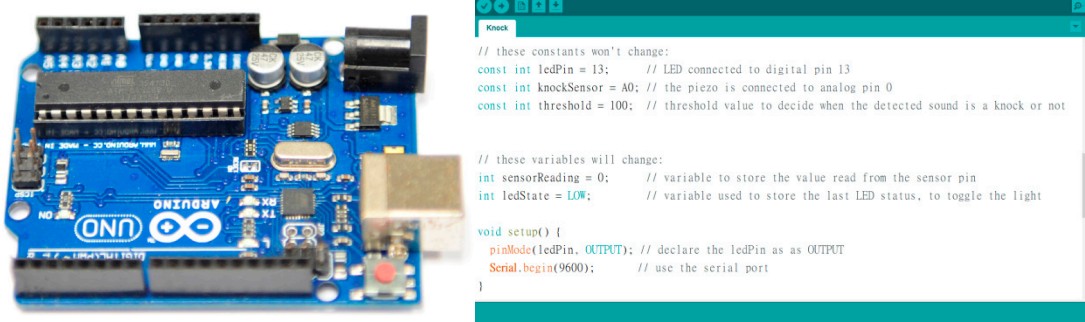

**Figure 1.** Arduino Uno control board and the screenshot of the editing software.

(2)     Webduino

Webduino (Figure 2) has been developed based on Arduino [68]. Arduino is based on C/C++, whereas Webduino is controlled using HTML and JavaScript and operated through Wi-Fi, WebSocket, Bluetooth, and a serial port.

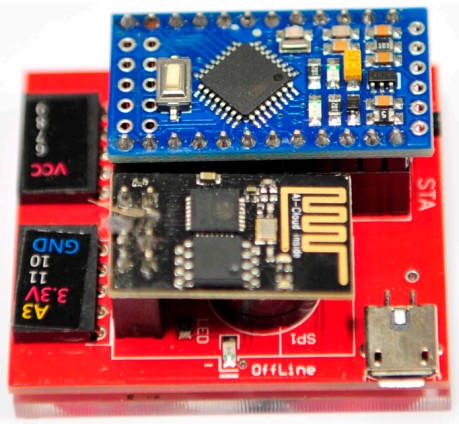

**Figure 2.** Webduino control board.

The name Webduino is a combination between Web and Arduino. Webduino allows its users to execute control actions by simply using syntax for websites; it enables users to connect Arduino to the Internet, update firmware in the cloud, write codes using various programming languages, and execute control actions using Wi-Fi. Because Webduino involves the use of web development technology, which facilitates cross-platform and cross-device operations, programs developed using Webduino show applicability beyond the conventionally used Arduino development boards to other development boards such as Raspberry Pi and esp8266. Numerous engineers have engaged in the development of Webduino to realize the true value and meaning of the IoT.

(3)    Flame detector

A flame detector (Figure 3) is sensitive to flames and also reactive to normal light. It is generally used for fire alarms and fire source detection and is able to detect flame or light with a wavelength ranging from 760 to 1100 nm. The maximum distance for detection of the flame of a lighter is 80 cm. The larger the flame, then the larger the maximum distance between the flame and the detector [69].

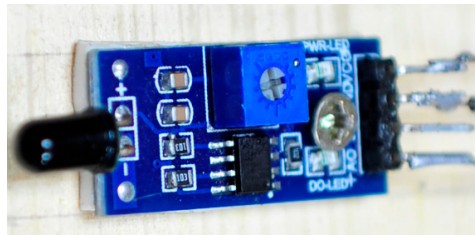

**Figure 3.** Flame detector.

(4)    DHT11 temperature sensor

The DHT11 digital temperature and humidity sensor (Figure 4) is a temperature and humidity composite sensor with a corrected digital signal output. It uses dedicated digital module acquisition technology and temperature and humidity sensing technology to ensure high reliability and excellent long-term stability. The product has the advantages of excellent quality, ultra-fast response, strong anti-interference ability and high cost performance. The single-wire connection makes the system combination easy and quick to install. Ultra-small size, extremely low power consumption, and signal transmission distances of up to 20 meters make it the best choice for applications. [70].

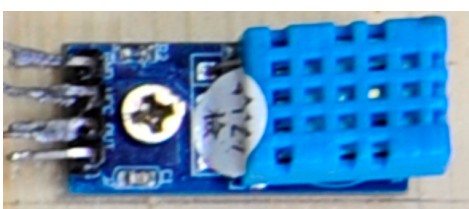

**Figure 4.** DHT11temperature sensor.

(5)    MQ-4 natural gas methane sensor

The gas-sensitive material used in the MQ-4 gas sensor (Figure 5) is $SnO_2$. When a combustible gas exists in an environment where the gas sensor is located, the conductivity of the sensor increases as the concentration of the combustible gas increases. The MQ-4 gas sensor is highly sensitive to methane as well as to propane and butane. It can detect numerous types of combustible gases, particularly natural gas, and thus is suitable for detecting methane gas and natural gas in households and factories. The detection range for natural gas and methane is from 300 to 10,000 ppm [71].

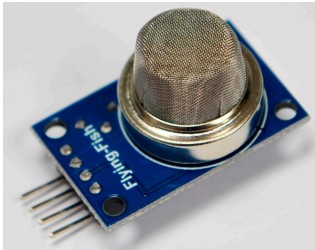

**Figure 5.** MQ-4 natural gas methane sensor.

(6)    IP camera

The camera employed in this study to monitor the gas stove is the 5th generation advanced version Yosef 1080P surveillance IP camera (Figure 6). It was installed at a height in the kitchen that allows for monitoring the situation of the gas stove; therefore, the users can monitor the gas stove from a remote distance [72].

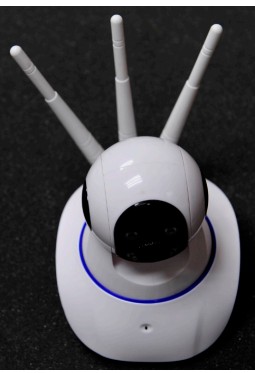

**Figure 6.** The Internet Protocol (IP) camera used in this study.

*3.2. The Smart Alarm and Control Mechanism*

In a smart building, various sensors are used to obtain environmental values at home, on the basis of which relevant control mechanisms are activated through the control board and software to issue notifications or switch on appliances to change the environment according to detected physical indicators, thereby constructing a safe and comfortable home environment. The devices used in this study to control the environment and issue warnings are as follows:

(1)    Relay

Relay (Figure 7) [73] is an electronic control device that has a control system (also known as an input circuit) and a controlled system (also known as an output circuit). It is typically used in automatic control circuits. It is an "automatic switch" that uses a small current to control a large current. Therefore, it plays the role of automatic adjustment, safety protection, conversion circuit, etc., in the circuit.

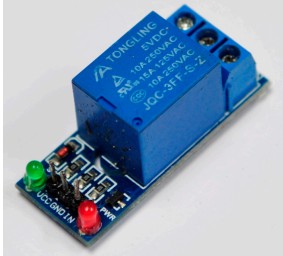

**Figure 7.** Relay.

(2) Display

This study used a 1602/16x2 monochrome character liquid–crystal display (LCD) module (Figure 8) to display the gas concentration, temperature, and flame intensity [74].

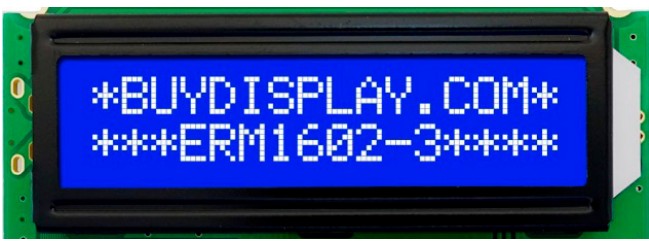

**Figure 8.** Liquid–crystal display (LCD) module.

(3) Sound and light alarm

If the sensor detects flames, high temperatures, or gas, then the sound and light alarm (Figure 9) produces an alarm sound and flashes to notify residents to adopt appropriate response measures.

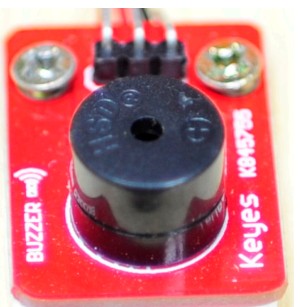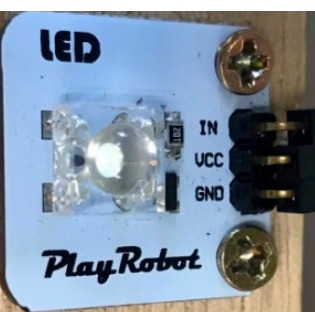

**Figure 9.** Sound and light alarm.

(4) Gas shutoff device

When the sensor detects flames, high temperature, or gas, the fire prevention system immediately activates the gas shutoff device (Figure 10). The valve is turned 90 degrees clockwise to turn off the gas supply, thereby ensuring the residents' safety [75].

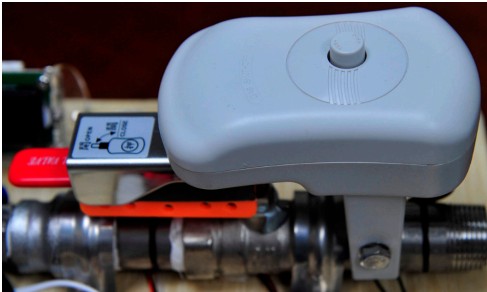

**Figure 10.** Gas shutoff device.

(5)    IFTTT

If This Then That (IFTTT) [76] is an automation tool that integrates many common web services. Users can create a conditional statement specifying that when a condition in Channel A is satisfied, it will trigger an action in Channel B; the entire process is a recipe. Currently, IFTTT is a free web service, but users must register an account with their e-mail address.

(6)    Line reporting system

Line [77] has been a popular mobile app in recent years because of its combination of communication and online services and the convenience of group chats in addition to individual chats. It not only can be installed on smartphones (e.g., iPhone, Android phones, Windows phones, Black Berry, and Nokia Asha) but can also be used on personal computers. Line facilitates the transmission of text, pictures, and videos and topic discussions in groups, serving as an extensively used tool for social networking, communication, and management. In this study, the IFTTT platform was used to obtain the data detected by various sensors in the smart building. The data were then sent to relevant Line users or groups in a timely manner (Figure 11).

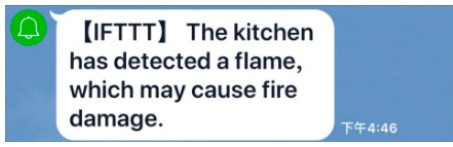

**Figure 11.** Notification messages on Line.

(7)    Door lock

When the sensor detects flames, high temperatures, or gas, the system automatically unlocks the main entrance door (Figure 12) to enable community management team and emergency personnel to enter the house to extinguish the fire.

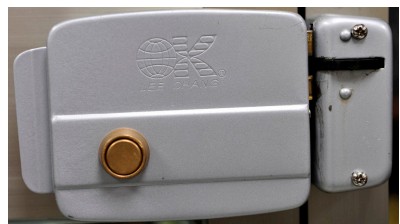

**Figure 12.** Door lock that can be controlled automatically and manually.

The brand, name, model, function quantity and price of hardware used in this study are listed on Table 2. The total cost of the materials used on this study is US$188.6.

**Table 2.** The hardware used in this study.

| Name | Brand/Model | Function | Price (US$) |
|---|---|---|---|
| Arduino | Uno | Detects sensor signals and initiates controlled facilities | 6.0 |
| Webduino | Mark1 | Remote control facility, such as shutting off gas supply | 28.0 |
| D1 Mini | Esp8266 | Transfer IFTTT and Line information | 3.0 |
| Flame detector | CSE0005-1 | Detect flame intensity | 0.5 |
| Gas methane sensor | MQ-4 CSS0013 | Detecting gas concentration | 2.0 |
| Temperature sensor | DHT11 | Detecting the temperature above the gas stove | 1.6 |
| IP camera | Yoose/1080P | Remote monitoring for kitchen fire usage | 23.0 |
| Display | Winstar1602/16x2 | Display detection values of the environment | 3.0 |
| LED alarm | | Generate alert flash to warn residents | 3.5 |
| Audible alarm | K 845755 | Generate alarm sounds to warn residents | 1.0 |
| Relay (3 pieces) | | Automatic switch | 3.0 |
| Gas shutoff device | Sunwe/AD-704T | Automatically cut off, manually recover | 75.0 |
| Door lock | Bird | Can be automatically opened by the system | 29.0 |
| Miscellaneous hardware | | Wiring and other materials | 10.0 |
| Total | | | 188.6 |

Software for the remote control of gas stove.

This study employed Webduino to develop remote control software. The software was connected with the IP camera through Wi-Fi to enable users to monitor the gas stove. The users can remotely control the gas shutoff device by simply clicking on a button to turn off the gas supply.

## 4. Results

The architecture of the smart kitchen fire prevention system constructed in this study is displayed in Figure 13.

In this study, the Arduino, Webduino and D1 Mini are used as the control panels (the middle part of Figure 13). The environmental data is collected by the gas detector, temperature detector, and flame detector (the upper part of Figure 13). The measured values are passed to Arduino and shown on the display. The editor and Blockly on the right of the figure is the program editing software of Arduino and Webduino respectively. The mobile phone can be connected to the IP Camera to check the cooking fire condition of the kitchen. When you find that you have not turned the stove off, you can connect to the Webduino by pushing the phone button and activate the relay to turn off the main gas switch. When any of the sensors detect any value exceeding the set value, the controllable mechanism on the bottom of the figure can be activated to trigger the alarm, open the door lock, send the Line notification message, as well as turn off the main gas switch. This study employed IoT to develop a smart kitchen fire prevention system. Creating the system architecture entailed integrating the control board, sensing elements, and controlled devices and developing control software. The setting of the controlled devices and Line notifications was established. The proposed system can serve as a reference for designing kitchen fire prevention systems.

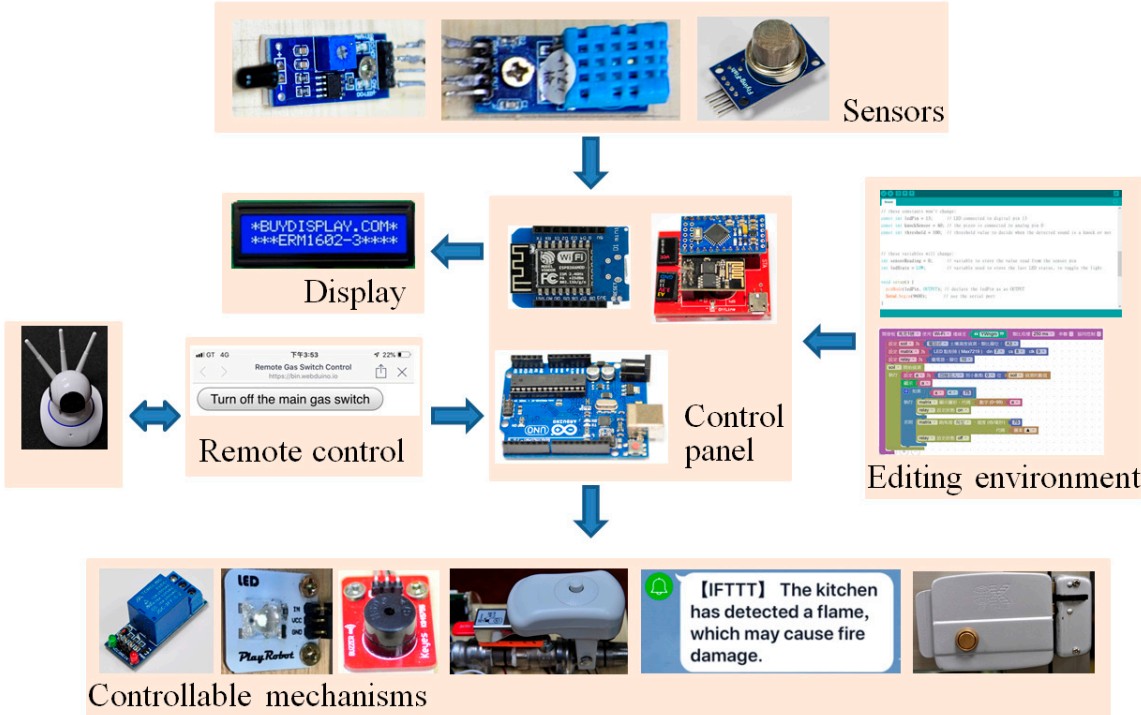

**Figure 13.** Architecture of the smart kitchen fire prevention system.

In this study, three control panels, Arduino, Webduino and D1 Mini, were used. Arduino sets up the connection and reads the sensor's detection data. It connects to the D1 Mini for notification, and activates the sound and flash alarm facility. It is also connected to the electrical appliance to open the door and close the gas main switch. These jobs do not require the use of Wi-Fi. This study uses Webduino to perform related remote invalidation controls. Because Webduino itself already has Wi-Fi communication capabilities, it can be connected to the network with a simple setup.

In this study, the MQ-4 gas sensor was used. We used the gas produced from a lighter to simulate the liquefied petroleum gas or natural gas to trigger the gas sensor. Hot air blown from a hair dryer was used to simulate a high temperature to trigger the temperature sensor, and a mobile phone flashlight was used to simulate a strong light to trigger the flame detector.

After the connection and testing of sensors, control board, and controlled devices were completed, we triggered the gas sensor, temperature sensor, and flame detector by using gas from the lighter, hot air from the hair dryer, and strong light from the mobile phone flashlight, respectively. The detected changes of gas concentration, temperature, and light intensity shown on the LCD are presented in Figure 14. When the gas concentration, temperature, and light intensity reached the set threshold, the alarm device emitted a warning light and sound (Figure 15), the gas shutoff device was activated to turn 90 degrees to turn off the gas supply (Figure 16), Line notification messages were sent to the user's mobile phone (Figure 17), and the main entrance door was automatically unlocked (Figure 18). When the system reads that any detected value exceeds the safe range, then the warning system, Line notification, opening the door lock and closing the gas will be triggered immediately. The automatic gas shut-off device takes about 5 s to completely cut off the gas supply.

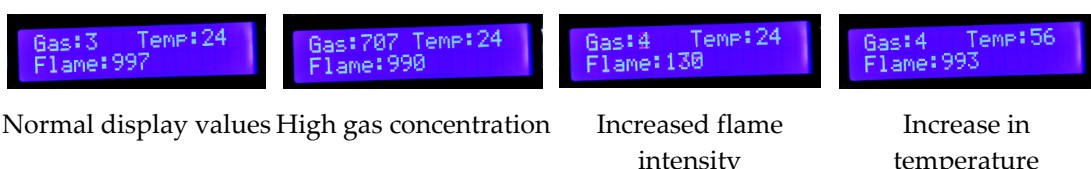

| Normal display values | High gas concentration | Increased flame intensity | Increase in temperature |

**Figure 14.** Changes of gas concentration, light intensity and temperature.

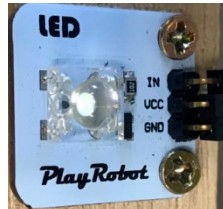
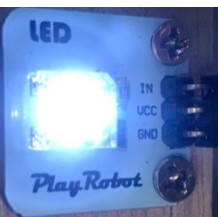

Alarm before activation                    Activated alarm

**Figure 15.** Activation of the alarm device.

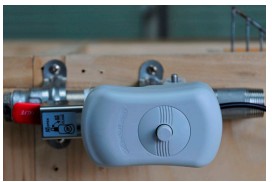
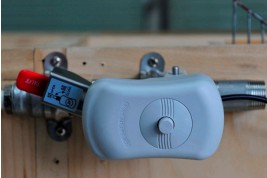
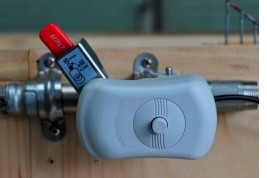
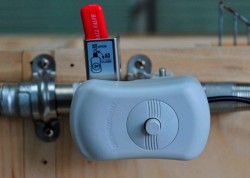

Gas completely ventilated | Shutoff device activated | Gas supply being cut off | Gas supply completely cut off

**Figure 16.** Gas shutoff device turned 90 degrees to turn off the gas supply.

【IFTTT】 The kitchen has detected high temperature, which may cause fire damage. 下午4:44 | 【IFTTT】 The kitchen has detected a flame, which may cause fire damage. 下午4:46 | 【IFTTT】 The kitchen has detected gas, which could be dangerous. 下午4:47

High temperature notification          Flame notification          Gas notification

**Figure 17.** Line notification messages on the mobile device.

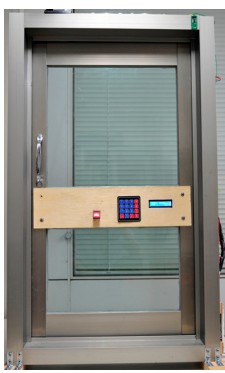
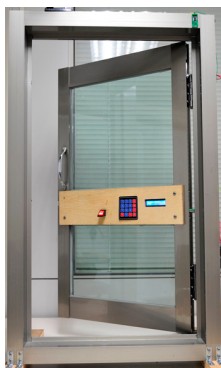

Front door locked                    Front door unlocked

**Figure 18.** Door being automatically unlocked.

This study has mainly used three kinds of sensors to detect the possible fire in the kitchen. When any of these sensors exceeds our set range, the relevant alarms, notifications, gas shuts and door locks are immediately activated. Their startup values, response time and completion/ended time are shown in Table 3. When the detected values reach the startup values, the system accurately starts the relevant facilities all the time.

**Table 3.** The startup value, response time and complete time of sensors/devices used in this study.

| Sensor/Device | Startup Value | Response Time | Complete/Ended Time |
|---|---|---|---|
| Flame detector | ≤600 | Immediately | N/A |
| Gas methane sensor | ≥500 | Immediately | N/A |
| Temperature sensor | ≥50 °C | Immediately | N/A |
| Sound and light alarm | N/A | Immediately | Detected values below/above startup values |
| Line reporting system | N/A | 1–2 s | Detected values below/above startup values |
| Gas shutoff device | N/A | Immediately | 4–5 s, manually open |
| Front door | N/A | Immediately | N/A, manually lock |

Regarding the remote monitoring of the gas stove, the IP camera was turned on to check whether the stove had been turned off. If not, the user clicked on the "Turn off the main switch" button (Figure 19) to activate the gas shutoff device and turn off the gas supply (Figure 16). Figure 20 shows images of the gas stove before and after the gas was shut off.

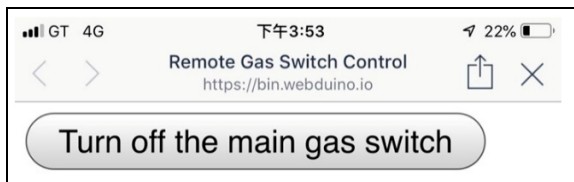

**Figure 19.** Interface for remotely turning off the gas supply.

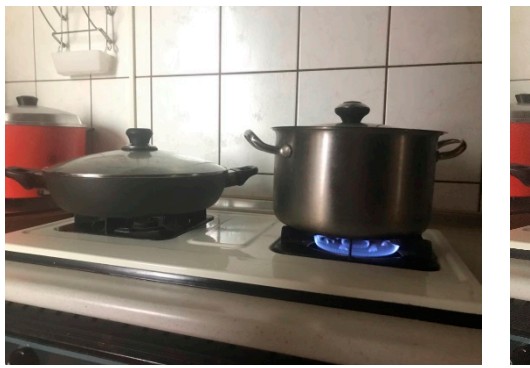 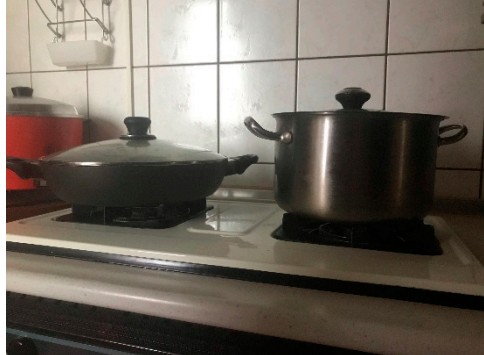

(**a**) Before gas shutoff         (**b**) After gas shutoff

**Figure 20.** Interface for remotely turning off the gas supply.

## 5. Discussion

This study originally adopted the normally open solenoid valve for the gas shutoff device. However, this type of valve must be continuously powered on to remain closed, which is unsuitable for use in the shutting off of the gas supply, because gas would continue to be supplied when the valve is not powered, resulting in danger. To completely shut off the gas supply, this study adopted the AD-704T gas shutoff device developed by Sunwe Technology. When the sensors detect flames, high temperatures, or a gas leak, the shutoff device is automatically turned 90 degrees clockwise to shut off the gas supply, ensuring residents' safety.

The traditional point-type smoke detectors have the disadvantage of not being able to effectively exert their fire detection effects in buildings with large areas and high floors [20,52]. The biggest drawback of video fire detectors is that these cannot detect early fires in time. This makes it impossible for the system to perform the most effective fire extinguishing function in the early stages of the fire. The flame sensor and temperature sensor used in this study will immediately trigger the sound and

light alarm, transmit the notification messages, cut off the gas supply and open the door lock when the flame intensity or temperature reaches the set value, so as to effectively serve the function of kitchen fire prevention and control.

When the detected values reach the set values in program, the system starts the relevant facilities immediately. It takes 1–2 s to transfer the Line notification. The gas shutoff device takes 4–5 s to complete the turning off of the gas.

While the kitchen fire prevention system in this study has sensors and a gas shutoff device, which is the same as the patent TW M501536, it also includes an alarm device, an automatic door unlocking function, and a Line message reporting system, thereby rendering the proposed kitchen fire prevention system more comprehensive.

This study's system is also similar to the patent TW 201709157 insofar as both systems have an automatic gas shutoff device, a wireless reporting function, and an alarm device. However, the proposed system further includes the functions of the detection of flame, temperature, and gas and automatic door unlocking, which enable the community management team and rescue personnel to immediately enter the house to fight the fire.

Our team has a lab space wherein the research results of the system have been displayed. There are many members of faculty, students and other users who have been able to experience the system and exchange opinions. Most of the friends and colleagues who have visited have given positive comments. In addition, our research team has organized the user opinions as a reference for future study and as potential direction for improvement of the system.

## 6. Conclusions

This study constructed a kitchen fire prevention system that can instantly activate an alarm device to issue warnings to the residents and shut off the gas supply when it detects a gas leak, flame, or high temperature in the kitchen. In addition, Line messages are sent to notify the community management team and emergency units and the main entrance door of the residence is automatically unlocked to allow the rescue personnel to enter the house to extinguish the fire. This can effectively reduce the annual casualties in kitchen fire accidents. In addition, the remote monitoring function enables users to monitor the gas stove at any time and turn off the stove by simply clicking on a button on their mobile phone. Future research directions can be combined with prevention of carbon monoxide poisoning to effectively protect the home and the safety of the residents.

**Author Contributions:** Conceptualization, Y.-C.S.; Funding acquisition, C.-K.L.; Investigation, W.-L.H.; Methodology, C.-S.H.; Project administration, W.-L.H.; Resources, J.-Y.J.; Software, C.-S.H.; Supervision, Y.-C.S.; Validation, J.-Y.J.; Visualization, C.-K.L.; Writing—original draft, Y.-C.S.; Writing—review and editing, W.-L.H.

**Funding:** This research received no external funding.

**Acknowledgments:** Our deepest gratitude goes to the anonymous reviewers for their careful work and thoughtful suggestions that have helped improve this paper substantially.

**Conflicts of Interest:** The author declares no conflict of interest.

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
