# Peer review of "Application of Internet of Things in a Kitchen Fire Prevention System"

_applsci, doi:10.3390/app9173520_

Round 1

Reviewer 1 Report

In general it is a nice, simple, practical and applicable useful paper

I have some notes as below:

The cost of the system maybe useful to be mentioned. 

Using the camera to check the flame picture is good but maybe adding extra cost to the system and we have the IFTTT notification system to get the real time notifications.

I think more details should be added about the Wi-Fi board that is connected to the Arduino.

Artificial Intelligence (AI) is mentioned and it needs to be discussed in more details and how it was useful for this paper and how could be related to human prediction. 

I think the steps to setup the IFTTT is obvious and no need to be mentioned in the paper.  

line 353 Missing is (It not only) should become (It is not only).  

Author Response

Reviewer-1: Comments and Suggestions for Authors

In general it is a nice, simple, practical and applicable useful paper.

I would like to show my appreciation to Reviewer-1 for spending so much time and providing a lot of good valuable comments, which will greatly help to improve the quality of our articles. The modified part of the article that responds to your suggestion is marked in blue. Thank you very much again.

I have some notes as below:

The cost of the system maybe useful to be mentioned. 

Response: Thank you. The component, function and cost of the system are listed on Table 2. (Lines 416 - 419)

Using the camera to check the flame picture is good but maybe adding extra cost to the system and we have the IFTTT notification system to get the real time notifications.

Response: The notification function of IFTTT and Line will be activated by D1 Mini only when the kitchen sensor detects high temperature, flame or gas. People often go out in a hurry and forget to bring things or turn off devices such as air-conditioning. After leaving the home for a long time and distance, one may wondered if he/she had turned off the cooking fire in kitchen. If there is no other people at home, one can use mobile phone connecting to the IP camera to confirm the usage of the cooking fire. If necessary, one can use the mobile phone to execute the software developed by this research and remotely turn off the main gas switch through the Webduino control board.

I think more details should be added about the Wi-Fi board that is connected to the Arduino.

Response: There are four common methods of IoT connection control (Lines 227 - 231). In this study, three control panels, Arduino, Webduino and D1 Mini, were used. Arduino sets up the connection and reads the sensor's detection data. It connects to the D1 Mini for notification, and activates the sound and flash alarm facility. It is also connected to the electrical appliance to open the door and close the main gas switch. These jobs do not require the use of Wi-Fi. This study uses Webduino to perform related remote invalidation controls. Because Webduino itself already has Wi-Fi communication capabilities, it can be connected to the network with a simple setup.

Artificial Intelligence (AI) is mentioned and it needs to be discussed in more details and how it was useful for this paper and how could be related to human prediction. 

Response: After your reminder, we found that this study only used the Internet of Things and did not use artificial intelligence. We removed the content related to artificial intelligence, including the title of this article (Lines 2 – 3 in this script), Section 2.3 (Lines 206 - 223) and Line 390 in the original text (applsci-538640.doc).

I think the steps to setup the IFTTT is obvious and no need to be mentioned in the paper.  

Response: The mentioned content has been deleted. (Lines 358 – 372 in applsci-538640.doc)

line 353 Missing is (It not only) should become (It is not only).  

Response: The mentioned error has been modified. (Line 402)

Reviewer 2 Report

The article presents a system for helping to prevent the spread of fire related to cooking activities. The system uses flame, temperature and gas sensors, as well as a camera and actuator to control the opening of the gas. 

The "literature review" section is more like a "background" section than a literature review. It focuses mainly on the definition of concepts and fire management and presents only two works related to the problem, which are two patents. This section needs to be expanded. Other work on the prevention/management of domestic accidents related to cooking exists, for example:

Bouchard, B., Bouchard, K., & Bouzouane, A. (2014, June). A smart range helping cognitively-impaired persons cooking. In Twenty-Sixth IAAI Conference.

Teslyuk, V., Beregovskyi, V., Denysyuk, P., Teslyuk, T., & Lozynskyi, A. (2018). Development and implementation of the technical accident prevention subsystem for the smart home system. International Journal of Intelligent Systems and Applications, 11(1), 1.

Saeed, F., Paul, A., Rehman, A., Hong, W., & Seo, H. (2018). IoT-based intelligent modeling of smart home environment for fire prevention and safety. Journal of Sensor and Actuator Networks, 7(1), 11.

The state of the art and positioning compared to all existing work must be done.

The Research design section has rather trivial sections (for example, the part on Line can be shortened). A section is missing to describe the algorithm used for the AI part. The system has remote control, but I wonder about that. Wouldn't the system be safer to automatically shut off the gas if it detected a risk rather than allow the user to remotely check from a video and put the control on?

The "result" part can be improved by describing the experimental protocol, hypotheses and results in a clearer way. It is a proof of concept, but is there any plan to conduct an experiment to measure the impact of this system on users? In particular, do they feel safer? Are they reluctant? etc. There is a lack of information on the behaviour of the system, including reaction times to different events.

Author Response

Reviewer-2: Comments and Suggestions for Authors

The article presents a system for helping to prevent the spread of fire related to cooking activities. The system uses flame, temperature and gas sensors, as well as a camera and actuator to control the opening of the gas.

I would like to express my gratitude for the many constructive suggestions you have proposed. The modified part of the article that responds to your suggestion is marked in red. Thank you very much again. 

The "literature review" section is more like a "background" section than a literature review. It focuses mainly on the definition of concepts and fire management and presents only two works related to the problem, which are two patents. This section needs to be expanded. Other work on the prevention/management of domestic accidents related to cooking exists, for example:

Bouchard, B., Bouchard, K., & Bouzouane, A. (2014, June). A smart range helping cognitively-impaired persons cooking. In Twenty-Sixth IAAI Conference.

Teslyuk, V., Beregovskyi, V., Denysyuk, P., Teslyuk, T., & Lozynskyi, A. (2018). Development and implementation of the technical accident prevention subsystem for the smart home system. International Journal of Intelligent Systems and Applications, 11(1), 1.

Saeed, F., Paul, A., Rehman, A., Hong, W., & Seo, H. (2018). IoT-based intelligent modeling of smart home environment for fire prevention and safety. Journal of Sensor and Actuator Networks, 7(1), 11.

Response: We have made a significant modification according your suggestion. The added materials include Lines 36 – 38, Lines 127 – 142, Lines 220 – 221, Lines 224 – 226, Lines 235 –239, Lines 244 – 246, Line 251, and Lines 258 – 262.

The new added references include References 1, 10, 11, 12, 13, 17, 18, 19, 20, 21, 23, 25, 28, 29, 30, 31, 32, 33, 34, 35, 36, 38, 39, 40, 41, 42, 43, 44, 45, 46, 47, 56.

The state of the art and positioning compared to all existing work must be done.

Response: The materials replying to mentioned suggestion have been added to Section 2.1 (Lines 110 – 125)

The Research design section has rather trivial sections (for example, the part on Line can be shortened). A section is missing to describe the algorithm used for the AI part. The system has remote control, but I wonder about that. Wouldn't the system be safer to automatically shut off the gas if it detected a risk rather than allow the user to remotely check from a video and put the control on?

Response: The part on Line can be shortened.

The mentioned content has been deleted. (Lines 358 – 372 in applsci-538640.doc)

A section is missing to describe the algorithm used for the AI part.

After your reminder, we found that this study only used the Internet of Things and did not use artificial intelligence. We removed the content related to artificial intelligence, including the title of this article (Lines 2 – 3 in this script), Section 2.3 (Lines 206 - 223) and Line 390 in the original text (applsci-538640.doc).

The system has remote control, but I wonder about that. Wouldn't the system be safer to automatically shut off the gas if it detected a risk rather than allow the user to remotely check from a video and put the control on?

Our explanation for your suggestion is as follows:

A.     When our sensors detect high temperature, flame or gas leakage, the system will automatically turn off the main gas switch.

B.     Our philosophy is that when people have been out for a while and have been away from home, they are not sure if the kitchen cooking fire is turned off. At this time, they can check the usage of the cooking fire by connecting the IP camera to mobile phone.

C.     If the cooking fire is not turned off, the main gas switch can be turned off using the remote control mechanism developed in this study.

D.     This type of processing is much safer than waiting until the system detects a fire hazard.

The "result" part can be improved by describing the experimental protocol, hypotheses and results in a clearer way. It is a proof of concept, but is there any plan to conduct an experiment to measure the impact of this system on users? In particular, do they feel safer? Are they reluctant? etc. There is a lack of information on the behaviour of the system, including reaction times to different events.

Response: The "result" part can be improved by describing the experimental protocol, hypotheses and results in a clearer way.

The responding message has been added on Lines 429 – 438.

It is a proof of concept, but is there any plan to conduct an experiment to measure the impact of this system on users? In particular, do they feel safer? Are they reluctant?

At the beginning of this study, the male-female jack type wiring was often unable to operate normally due to poor contact. Later, this study used the welding method for the line connection, and never had a false alarm, accidentally opened the door lock, or mis-closed the gas switch. This shows that the stability of the system is quite high. The partial responding message has been added on Lines 501 – 504.

There is a lack of information on the behaviour of the system, including reaction times to different events.

The responding message has been added on Lines 461 – 464.

Round 2

Reviewer 2 Report

A significant improvement has been done to the paper and the title seems better suited to the proposal.

The literature review section does not present all the work related to fire prevention in a kitchen. This section is much more like a presentation of the prerequisites, but not a complete review of the related work. Section 2.4 presents 2 patents that compare the work with. However, works such as those listed above are omitted without explanation (partial list).

- Mobin, M. I., Abid-Ar-Rafi, M., Islam, M. N., & Hasan, M. R. (2016). An intelligent fire detection and mitigation system safe from fire (SFF). Int. J. Comput. Appl133(6), 1-7.

- Abdulrazak, B., Yared, R., Tessier, T., & Mabilleau, P. (2015, May). Toward Pervasive Computing System to Enhance Safety of Ageing People in Smart Kitchen. In ICT4AgeingWell (pp. 17-28).

- Dukuzumuremyi, J. P., Zou, B., & Hanyurwimfura, D. (2014). A novel algorithm for fire/smoke detection based on computer vision. International Journal of Hybrid Information Technology7(3), 143-154. (This article is an example but why exclude vision-based fire detection approaches?)

- Apeh, S. T., Erameh, K. B., & Iruansi, U. (2014). Design and Development of Kitchen Gas Leakage Detection and Automatic Gas Shut off System. Journal of Emerging Trends in Engineering and Applied Sciences (JETEAS)5(3), 222-228. (This work deals with certain aspects of your work)

A presentation of the research criteria and analysis criteria would facilitate the evaluation of the contribution. In my opinion, section 2.1 to 2.3 could be restructured into a section called background or prerequisite. Section 2.4 is the "real" literature review.

The addition of the total cost of the solution is a good point and seems to me a good factor to compare with other work.

The result section could be improved in terms of structure by presenting for each scenario the state of the system at the beginning, the action taken and the reaction of the system. I think it would also be very interesting in this context to have reaction times (this criterion could also be an element of comparison with the work of the literature). In addition, the authors indicate in their response that manual gas shutdown by the user using the remote application can be safer (because it is faster according to my understanding) than the detection of a fire risk by the system.

If we add the above-mentioned criteria of comparison, the discussion section will also improve by criticizing the system's behaviour.

I also ask the question, could the system not adapt without too much difficulty to an electric stove?

Author Response

Comments and Suggestions for Authors

A significant improvement has been done to the paper and the title seems better suited to the proposal. 

I would like to show my appreciation to Reviewer-2 for spending so much time and providing a lot of good valuable comments, which will greatly help to improve the quality of our articles. The modified part of the article that responds to your suggestion is marked in red. Thank you very much again.

The literature review section does not present all the work related to fire prevention in a kitchen. This section is much more like a presentation of the prerequisites, but not a complete review of the related work. Section 2.4 presents 2 patents that compare the work with. However, works such as those listed above are omitted without explanation (partial list). 

- Mobin, M. I., Abid-Ar-Rafi, M., Islam, M. N., & Hasan, M. R. (2016). An intelligent fire detection and mitigation system safe from fire (SFF). Int. J. Comput. Appl133(6), 1-7.

- Abdulrazak, B., Yared, R., Tessier, T., & Mabilleau, P. (2015, May). Toward Pervasive Computing System to Enhance Safety of Ageing People in Smart Kitchen. In ICT4AgeingWell (pp. 17-28).

- Dukuzumuremyi, J. P., Zou, B., & Hanyurwimfura, D. (2014). A novel algorithm for fire/smoke detection based on computer vision. International Journal of Hybrid Information Technology7(3), 143-154. (This article is an example but why exclude vision-based fire detection approaches?)

- Apeh, S. T., Erameh, K. B., & Iruansi, U. (2014). Design and Development of Kitchen Gas Leakage Detection and Automatic Gas Shut off System. Journal of Emerging Trends in Engineering and Applied Sciences (JETEAS)5(3), 222-228. (This work deals with certain aspects of your work)

Response: I would like to show my appreciation for your precious suggestion. The above and some other literature review have been added on Section 2.4 (Lines 264 -324).

A presentation of the research criteria and analysis criteria would facilitate the evaluation of the contribution. In my opinion, section 2.1 to 2.3 could be restructured into a section called background or prerequisite. Section 2.4 is the "real" literature review. 

Response: Thank you for making such constructive suggestions. However, after discussing with other authors, we feel that the review section of the academic research you proposed is placed in Sections 2.4.1 - 2.4.2, and the review section of the patent is placed in 2.4.3. This arrangement is more logical and has minimal impact on the structure of the original article. But if you still think that your approach is appropriate, we will make adjustments based on your proposal.

The addition of the total cost of the solution is a good point and seems to me a good factor to compare with other work.

Response: Thank you for your recognition and encouragement.

The result section could be improved in terms of structure by presenting for each scenario the state of the system at the beginning, the action taken and the reaction of the system. I think it would also be very interesting in this context to have reaction times (this criterion could also be an element of comparison with the work of the literature).

Response: Thank you for your recommendation. The system reaction scenarios have been discussed on Lines 518 – 522.

I have added the discussion on Section 5 (Discussion, Lines 553 - 560) for reaction times of different detection system referring in Section 4.2.

In addition, the authors indicate in their response that manual gas shutdown by the user using the remote application can be safer (because it is faster according to my understanding) than the detection of a fire risk by the system.

Response: Thank you for your recognition and encouragement.

If we add the above-mentioned criteria of comparison, the discussion section will also improve by criticizing the system's behaviour.

Response: Thank you. The mention comparison has been added in Section 5 (Lines 553 - 560).

I also ask the question, could the system not adapt without too much difficulty to an electric stove?

Response: According to my understanding, gas heating cooking and electric stove heating cooking both may cause fire due to overheating of the soup.

In the entire fire detection and disaster prevention process, except for the gas detector that is not used in the electric stove system, the rest can be used without any changes.

Round 3

Reviewer 2 Report

The article has definitely improved in quality, although there is still some way to improve it.

My main comments concern literature review. It would be nice to have an explanation of the methodology used to select articles, such as the criteria for selection and exclusion of articles. The authors explain that some articles were excluded in their letter, but without providing any explanation.

The result section can be improved in its form by better highlighting the protocol, the elements observed (response time, correct detection rate, etc.) and how they are observed. A summary table can be a plus.

The discussion section can be further improved by discussing other criteria such as system response times or fire start recognition rates for example.

Finally, I also have some minor remarks:

Some sentences should be supported by references. For example, lines 284 and 553-554.

The font of the title of Figure 2 is strange.

In section 2.4.3, the titles (TW M501536 and TW 201709157) should be changed to reflect the names of the patents and not their identifiers.

There are some references that are incorrectly positioned in the text (e.g. lines 268, 270, 275, 278, 287).

I have some difficulties with the sentences in lines 299 to 301 (it looks a bit like a repetition) and line 309.

Author Response

The article has definitely improved in quality, although there is still some way to improve it.

Thank you very much for your help. It took you so long, so detailed to us to modify our articles, so that our research can have a more complete presentation.

My main comments concern literature review. It would be nice to have an explanation of the methodology used to select articles, such as the criteria for selection and exclusion of articles. The authors explain that some articles were excluded in their letter, but without providing any explanation.

Response: As my understanding, you have suggested 3 papers in 1st review and 4 articles in 2nd review. These papers all have been in our script as references. Thank you for your kindly help.

1st Review:

Bouchard, B., Bouchard, K., & Bouzouane, A. (2014, June). A smart range helping cognitively-impaired persons In Twenty-Sixth IAAI Conference.--- Reference [10] Teslyuk, V., Beregovskyi, V., Denysyuk, P., Teslyuk, T., & Lozynskyi, A. (2018). Development and implementation of the technical accident prevention subsystem for the smart home system. International Journal of Intelligent Systems and Applications, 11(1), 1. Reference [20] Saeed, F., Paul, A., Rehman, A., Hong, W., & Seo, H. (2018). IoT-based intelligent modeling of smart home environment for fire prevention and safety. Journal of Sensor and Actuator Networks, 7(1), 11. Reference [1]

2nd Review:

- Mobin, M. I., Abid-Ar-Rafi, M., Islam, M. N., & Hasan, M. R. (2016). An intelligent fire detection and mitigation system safe from fire (SFF). Int. J. Comput. Appl, 133(6), 1-7. Reference [51] - Abdulrazak, B., Yared, R., Tessier, T., & Mabilleau, P. (2015, May). Toward Pervasive Computing System to Enhance Safety of Ageing People in Smart Kitchen. In ICT4AgeingWell (pp. 17-28). Reference [49] - Dukuzumuremyi, J. P., Zou, B., & Hanyurwimfura, D. (2014). A novel algorithm for fire/smoke detection based on computer vision. International Journal of Hybrid Information Technology, 7(3), 143-154. (This article is an example but why exclude vision-based fire detection approaches?) Reference [60] - Apeh, S. T., Erameh, K. B., & Iruansi, U. (2014). Design and Development of Kitchen Gas Leakage Detection and Automatic Gas Shut off System. Journal of Emerging Trends in Engineering and Applied Sciences (JETEAS), 5(3), 222-228. (This work deals with certain aspects of your work) Reference [58]

The result section can be improved in its form by better highlighting the protocol, the elements observed (response time, correct detection rate, etc.) and how they are observed. A summary table can be a plus.

Response: The title of this article is the kitchen fire prevention system. The article mainly uses three kinds of sensors to detect the possible fire in the kitchen. When any of these sensors exceeds our set range, the relevant alarms, notifications, gas shuts and door locks are immediately activated. Their startup values, response time and completion/ended time are shown in Table 3. When the detected values reach the startup values, the system accurately starts the relevant facilities all the time.

The discussion section can be further improved by discussing other criteria such as system response times or fire start recognition rates for example.

Response: The responding content has been added in Discussion section. (Lines 565 – 567)

Finally, I also have some minor remarks:

Some sentences should be supported by references. For example, lines 284 and 553-554.

Response: The responding references have been added in the paper. (Line 284, Line 558)

The font of the title of Figure 2 is strange.

Response: The font of the title in Figure 2 has been changed. Thank you. (Line 385)

In section 2.4.3, the titles (TW M501536 and TW 201709157) should be changed to reflect the names of the patents and not their identifiers.

Response: The titles of these patents has been added in Section 2.4.3. Thank you. (Line 325, Line 340)

There are some references that are incorrectly positioned in the text (e.g. lines 268, 270, 275, 278, 287).

Response: The stated problem has been fixed.

I have some difficulties with the sentences in lines 299 to 301 (it looks a bit like a repetition) and line 309.

Response: The repetition portion of the article has been removed.
